# Fibroblast–Myofibroblast Transition in Osteoarthritis Progression: Current Insights

**DOI:** 10.3390/ijms26167881

**Published:** 2025-08-15

**Authors:** Ruixin Peng, Qiyuan Lin, Zhen Yang, Hui Li, Jiao Jiao Li, Dan Xing

**Affiliations:** 1Arthritis Clinical and Research Center, Peking University People’s Hospital, No. 11 Xizhimen South Street, Beijing 100044, Chinaqiyuanlin_pku@163.com (Q.L.);; 2School of Biomedical Engineering, Faculty of Engineering and IT, University of Technology Sydney, Sydney, NSW 2007, Australia; jiaojiao.li@uts.edu.au

**Keywords:** fibroblast-like synoviocytes, osteoarthritis, synovial fibrosis, fibroblast–myofibroblast transition

## Abstract

Osteoarthritis (OA) is a multifactorial joint disease traditionally characterized by cartilage degradation, while growing evidence underscores the critical role of synovial fibrosis in driving disease progression. The synovium exhibits pathological remodeling in OA, primarily due to the phenotypic transition of fibroblast-like synoviocytes (FLSs) into myofibroblasts. This fibroblast–myofibroblast transition (FMT) results in excessive deposition of extracellular matrix (ECM) and increased tissue stiffness and contractility, collectively contributing to chronic inflammation and fibrotic stiffening of the joint capsule. These fibrotic changes not only impair synovial function but also exacerbate cartilage degeneration, nociceptive sensitization, and joint dysfunction, thereby amplifying OA severity. Focusing on the frequently overlooked role of the FMT of synovial fibroblasts in OA, this review introduces the biological characteristics of FLSs and myofibroblasts and systematically examines the key molecular pathways implicated in OA-related FMT, including TGF-β, Wnt/β-catenin, YAP/TAZ, and inflammatory signaling cascades. It also discusses emerging therapeutic strategies targeting synovial fibrosis and FMT and considers their implications for the clinical management of OA. By highlighting recent advances and unresolved challenges, this review provides critical insights into the fibroblast–myofibroblast axis as a central contributor to OA progression and a promising therapeutic target for modifying disease trajectory.

## 1. Introduction

### 1.1. Osteoarthritis

Osteoarthritis (OA) is the most prevalent form of joint disease globally, representing a major cause of disability and a growing burden on healthcare systems, which is mentioned by plenty of articles based on data from the GBD. As of 2021, there were roughly 595 million people suffering from OA, accounting for 7.6% of the world’s population. The prevalence of OA saw a surge of 132.2% in the last three decades, and this ascending trend is predicted to continue [1]. Hence, it is essential to increase concentration on the long-overlooked disease. While historically considered a disorder of cartilage degeneration, OA is now recognized as a whole-joint disease involving complex interactions among multiple tissues. In addition to hallmark features such as articular cartilage degradation, osteophyte formation, and subchondral bone remodeling, the synovium plays a crucial role in OA pathogenesis. Synovial alterations, including inflammatory infiltration and synovial fibrosis, are increasingly understood to actively contribute to joint dysfunction rather than merely reflect the downstream consequences of cartilage breakdown, as reviewed by Zhang et al. [2,3]. Clinically, these changes as well as other OA-contributing factors collectively manifest as joint stiffness, pain, swelling, and loss of mobility. The evolving “whole joint” paradigm has prompted a renewed understanding of synovial pathology, positioning synovitis and fibrosis not only as secondary features but also as potential drivers of OA progression. This shift underscores the importance of targeting synovial tissue responses in therapeutic strategies and increasing the current understanding of molecular and cellular mechanisms that govern fibrotic remodeling in the OA joint.

### 1.2. Fibroblast-like Synoviocytes: Definition, Features, and Heterogeneity

Fibroblast-like synoviocytes (FLSs) play a pivotal role in synovial homeostasis and contribute significantly to pathological changes within not only the synovium but the entire joint environment. FLSs are specialized mesenchymal cells within the synovial membrane originally described as spindle-shaped fibroblasts embedded in a collagen-rich extracellular matrix (ECM), as first observed by Barland et al. under an electron microscope [4,5]. As part of their normal physiological function, FLSs secrete key lubricating molecules such as hyaluronic acid and proteoglycan 4 (PRG4), which provide nutritional support for the articular cartilage and reduce friction and support cartilage integrity during joint motion [6]. They also produce structural ECM proteins including collagen and fibronectin as well as regulate their cross-linking, thereby influencing ECM organization and mechanical properties [2,6]. In relation to regulating matrix turnover, FLSs produce catabolic enzymes including matrix metalloproteinases (MMPs) and a disintegrin and metalloproteinases with thrombospondin motifs (ADAMTS) that degrade ECM components and maintain joint homeostasis. Thus, FLSs play central roles in shaping joint ECM components, structure, and physicochemical as well as mechanical properties, while these in turn regulate tissue-resident cell behavior such as proliferation, migration, and differentiation [7,8,9].

FLSs are functionally heterogeneous and can be classified into several subpopulations: Based on single-cell studies like scRNA-seq transcriptomics, FLSs mainly show two subtypes that differ in locations, gene expressions, as well as physiological functions [10,11]. CD55+ CD34− THY1− FLSs, situated in the synovial lining layer, express higher level of FN1, CD55, PRG4, HBEGF, and CLIC5 [12], relevant to cell adhesion, collagen layout, lubrication, and mitochondrial ROS regulation, respectively. This subpopulation also promotes osteogenesis through the expression of BMP-6 while it triggers osteoclastogenesis by expressing RANKL and CCL9 [11], which may affect the metabolism of bones. In comparison, the THY1+ FLS is found in the synovial sub-lining layer with various phenotypes. The group plays a positive role in pro-inflammatory activity and chronic synovitis such as angiogenesis [13]. Among the subsets of THY1+ FLS, the THY1+ CD34− DKK3+ HLA-DRlow FLS is of greater importance in the synovial fibrosis under OA context. Normally, this type of FLS produces more BAX so that apoptosis is enhanced. Moreover, the inhibition of the TGF-β/Smad signaling pathway leads to a reduction in collagen secretion [14,15].

Notably, pathological alterations in the synovial lining layer, including CD55+ CD34− THY1− FLS hyperplasia and phenotypic transformation, are particularly prominent in OA. Furthermore, THY1+ CD34− DKK3+ FLS in the sub-lining layer also experience a surge in OA circumstance. According to the research by Chen et al., DKK3+ FLS may be the major subcluster that differentiates to the myofibroblast [16]. The aberrant activation and dysfunction caused by transition of these FLS subtypes are likely to break cartilage and bone homeostasis, plus paving the way for synovial fibrosis.

### 1.3. OA-Associated FLS Changes and Synovial Fibrosis

In addition to their roles in tissue homeostasis, FLSs are central players in tissue repair following injury. Upon tissue damage, FLSs become activated and often differentiate into a myofibroblast phenotype. This transition is characterized by increased proliferation, enhanced secretion of ECM rich in type I and type III collagen, and the formation of contractile stress fibers containing α-smooth muscle actin (α-SMA). These stress fibers anchor to collagen fibrils through integrins, generating high contractile forces that contribute to the formation of structurally intact and mechanically stable but functionally impaired scar tissue. Under normal conditions, the transient and moderate activation of FLSs into myofibroblasts is beneficial for emergency repair of tissue injuries. Subsequently, the myofibroblasts undergo apoptosis or dedifferentiate, and the ECM is remodeled toward a normal tissue architecture. However, under pathological conditions, persistent fibroblast activation and impaired myofibroblast clearance lead to excessive ECM accumulation, synovial thickening, and joint swelling. This aberrant remodeling replaces functional tissue with fibrotic scarring, resulting in harmful fibrosis coupled with long-term structural and mechanical dysfunction. The distinct mechanisms determining transient versus persistent myofibroblast fate remain incompletely understood. It is thought that sustained exposure to specific mechanical cues (tissue strain, stiffness) and biochemical factors (growth factors, ECM ligands) induces an epigenetic memory, converting transient myofibroblasts into persistent ones [17,18]. Myofibroblasts expressing α-SMA have been detected in both the synovial lining and sub-lining layers of OA joints [14], highlighting their contributions across synovial compartments. In OA, the sustained activation of FLSs into myofibroblasts drives synovial fibrosis, which alters the composition and viscosity of synovial fluid, disrupts cartilage–ECM interactions, and compromises overall joint function.

Compared with other known cellular transitions or processes, we understand only a smattering of the fibroblast–myofibroblast transition (FMT) despite its significance and potential. Therefore, it is time to specifically focus on FMT and tackle its puzzles. Emerging evidence supports the view that synovial fibrosis is not merely a consequence of OA but a pathogenic mechanism that actively accelerates disease progression. Research by Yan et al. indicated that a drug specific for pulmonary fibrosis, nintedanib, remarkably mitigated articular cartilage degeneration plus reduced the quantity of M1 macrophages in a mouse OA model [19]. Similarly, a study by Wei et al. illustrated pirfenidone’s (a potent anti-fibrotic drug) effect on lowering articular cartilage degeneration as well as OARSI scores of the OA rabbit model, which means the progression of OA is deferred [20]. Synovial fibrosis derived from FMT plays a positive role in other OA pathological processes, including cartilaginous degeneration and synovial inflammation, despite requirements for further explanation of the concrete mechanism. Combating synovial fibrosis proved a worthy endeavor to alleviate OA progress. The potential interaction between FMTs and other cellular changes would help reveal the dynamic process of OA, better estimating the disease. Overall, paying attention to FMTs may provide some novel insights for understanding OA.

## 2. Molecular Pathways of Pathological FLS Activation

FLS activation into a myofibroblast-like state underlies the pathological remodeling of the synovium and drives many of the fibrotic and inflammatory features of OA progression. This phenotypic shift is orchestrated by a complex interplay of external stimuli and intracellular signaling pathways. Mechanical stress from altered joint loading, inflammatory cytokines from immune and resident stromal cells, and changes in cell autophagy all serve as potent inducing factors. These cues converge on several key molecular pathways, including TGF-β/Smad, Wnt/β-catenin, and Hippo-YAP/TAZ, which regulate FLS proliferation, differentiation, matrix production, and contractility. Understanding the molecular mechanisms driving pathological FLS activation is essential for identifying potential targets for anti-fibrotic and disease-modifying therapies in OA.

### 2.1. Inducing Factors

#### 2.1.1. Mechanical Stress

Pathological mechanical forces, including abnormal tension generated by myofibroblast contraction and fibrotic ECM, compressive loading, shear stress from joint motion, and hydrostatic pressure from synovial fluid, promote the differentiation of FLSs into myofibroblasts. These mechanical cues promote a feed-forward loop of fibrosis and inflammation. Specifically, mechanical stress increases the expression of TGF-β1, a key inducer of FLS activation and pro-inflammatory gene expression. Pathological stretching has also been shown to reduce lysyl oxidase levels, a critical enzyme for collagen and elastin crosslinking in the ECM [14]. Jamal et al. [21] reported that in early-stage OA, prior to overt joint degeneration, FLSs exposed to excessive mechanical tension exhibited significantly elevated COL1A1 mRNA expression compared with cells under physiological tension or from mid- to late-stage OA joints, suggesting that tension may initiate FLS activation and early fibrotic responses contributing to OA onset. In this study, primary patient-derived FLSs were subjected to biochemical stimulation through OA-conditioned media to create low- and high-inflammatory environments, alongside uniaxial stretch to confer aberrant mechanical tension. It was found that aberrant tension upregulated TGF-β1 in FLSs and acted in synergy with a high-inflammatory OA environment to significantly increase the expression of ADAMTS4 and ADAMTS5, the main proteoglycanases that degrade the matrix of articular cartilage, driving the development of an OA-like phenotype [21].

As a well-established initiating factor in OA, mechanical stress imbalance not only precipitates cartilage damage but also activates a secondary inflammatory response in both the cartilage and synovium. Due to excessive mechanical stress, chondrocytes and other broken cells release cartilage debris, cytokines, and growth factors which participate in the inflammatory cascade. Mechanical stress promotes the invasive and fibrogenic phenotype of FLSs both directly, as demonstrated above, and indirectly via the interaction with inflammatory signaling to perpetuate synovial inflammation [22,23,24]. This highlights its central role in driving synovial fibrosis and OA progression.

#### 2.1.2. Inflammatory Factors

As a low-grade inflammatory, degenerative, and pan-articular disease, OA has a more complex and less fully understood set of pathogenic mechanisms compared with other types of joint diseases. Chronic inflammation plays a key role in the progression of OA and is, to a significant extent, mediated by synovium-related changes. In the early stages of OA, synovitis is considered a key driver for FLS activation and subsequent synovial fibrosis driven by pro-inflammatory cytokines such as IL-1β, TNF-α, and IL-6 as well as chemokines like IL-8 (CXCL8) and MCP-1 (CCL2) secreted by polarized macrophages and senescent FLSs [5]. These inflammatory factors, on the one hand, directly promote the activation of FLSs into myofibroblasts. These inflammatory mediators contribute directly to the transdifferentiation of FLSs into myofibroblasts while also promoting angiogenesis and monocyte infiltration into the synovium, hence perpetuating joint inflammation and fibrosis through a self-reinforcing feedback loop.

Although the precise mechanisms of inflammation-induced fibrosis in OA remain to be fully elucidated, current evidence has suggested several important signaling pathways. IL-1β binds to its receptor IL-1RI, which can activate the MAPK (via ERK) and NF-κB pathways [25]. IL-6, in combination with the sIL-6R complex, binds to the gp130 protein and activates the JAK/STAT, PI3K, and MAPK signaling pathways [25,26,27]. IL-8 can enhance the phosphorylation of STAT3 and the NF-κB subunit p65 [28], while also acting as a chemoattractant to recruit neutrophils, thereby amplifying the inflammatory response and the secretion of other pro-inflammatory cytokines. TNF-α and IL-17 can induce Wnt5a expression, thereby activating the Wnt/β-catenin signaling pathway and increasing the expression of α-SMA and cadherin-11 in FLSs [14,29,30]. TNF-α also enhances a pro-inflammatory FLS phenotype through the NF-κB/AP-1 pathway. Additionally, IL-18 upregulates IL-6 and TNF-α expression, reinforcing the vicious cycle of inflammatory factor activation in FLSs [31]. Interestingly, while IL-4 is classically anti-inflammatory, it may also exert profibrotic effects by stimulating fibroblasts to produce type I and III collagen.

#### 2.1.3. Damage-Associated Molecular Patterns (DAMPs)

Under pathological conditions such as cellular stress or persistent tissue damage, damage-associated molecular patterns (DAMPs), which serve as endogenous danger signals, are released into the extracellular microenvironment [32]. These DAMPs bind to pattern recognition receptors (PRRs), which are widely expressed across innate immune and stromal cells. Key PRR families include Toll-like receptors (TLRs), NOD-like receptors (NLRs), the IL-1 receptor (IL-1R), and RIG-I-like receptors. Ligand binding to PRRs initiates the innate immune response, including the activation of the NLRP3 inflammasome. Upon sensing activation signals, NLRP3 undergoes oligomerization and recruits the adaptor protein ASC, which in turn binds pro-caspase-1 to form a functional complex. Within this active inflammasome complex, pro-caspase-1 is cleaved into caspase-1, its mature, enzymatically active form, through autocatalysis. Activated caspase-1 then processes pro-IL-1β and pro-IL-18 into their matured and activated forms, leading to the release of potent inflammatory cytokines. Ultimately, inflammasome activation leads to the release of IL-1β and may even induce apoptosis, further amplifying the release of inflammatory cytokines and intracellular DAMPs [33,34,35]. Through this mechanism, DAMPs indirectly promote FLS activation by sustaining and intensifying synovial inflammation. ECM fragments such as fibronectin, type II collagen, and cartilage oligomeric matrix protein (COMP), generated by enzymatic degradation through MMPs and ADAMTS or by mechanical damage to cartilage, also function as DAMPs and can activate FLSs through integrin- and TLR-mediated signaling pathways [6]. In addition, inorganic crystals such as hydroxyapatite, calcium phosphate, and monosodium urate in the joint can act as DAMPs that primarily activate the NLRP3 inflammasome through TLR4-dependent pathways [14], further linking tissue damage to fibrotic responses in OA.

#### 2.1.4. Abnormal Cell Autophagy

Trauma and other OA initiators can disrupt autophagy in FLSs, and this impairment contributes significantly to disease progression through two main mechanisms. First, impaired autophagy promotes the development of the senescence-associated secretory phenotype (SASP), characterized by the release of pro-inflammatory cytokines, particularly interleukins, which amplify inflammation and drive fibrotic responses. This pro-inflammatory environment also activates fibrogenic pathways, transforming FLSs into a myofibroblast phenotype, notably through TGF-β signaling pathways. Moreover, SASP-related secretions include molecules such as tissue inhibitors of metalloproteinases (TIMPs), which further contribute to ECM accumulation and synovial fibrosis [2]. Secondly, impaired autophagy leads to mitochondrial dysfunction and excessive production of reactive oxygen species (ROS), particularly mitochondrial ROS (mtROS), generated by NADPH oxidase activity and electron leakage from the mitochondrial electron transport chain [14,36]. This excessive mtROS serves as a key intermediate link in TGF-β signaling and is required for the upregulation of fibrotic markers such as α-SMA. Supporting this mechanism, Jain et al.’s study using lung fibroblasts showed that inhibiting ROS production from mitochondrial complex III attenuates the TGF-β-induced expression of pro-fibrotic genes and reduces the expression of NADPH oxidase 4, a key mediator of myofibroblast differentiation [37]. Collectively, these findings highlight autophagy dysfunction as a central driver of general myofibroblast differentiation in various organs [38,39]. We infer that effect would also apply to FLS activation and synovial fibrosis in OA, acting through both inflammatory and oxidative stress-dependent pathways. However, future studies confirming the specific role of ROS in FLS activation are needed (Figure 1).

### 2.2. Intracellular Signaling Pathways

#### 2.2.1. TGF-β/Smad Signaling Pathway

TGF-β is a ubiquitously expressed and multifunctional cytokine family comprising three isoforms: TGF-β1, TGF-β2, and TGF-β3. It plays a crucial role in regulating cellular processes such as survival, proliferation, differentiation, and immune modulation. Specifically, TGF-β is a central mediator for fibroblast-to-myofibroblast transition, tissue repair, and pathological fibrosis. The TGF-β precursor forms a dimer through disulfide bonds and is then proteolytically cleaved into two components: the mature TGF-β cytokine and the latency-associated peptide (LAP). These two components remain non-covalently associated to form the small latent complex (SLC), in which the receptor-binding site of TGF-β is sterically masked by LAP, thereby maintaining TGF-β in an inactive state. The SLC further associates with latent TGF-β binding proteins (LTBPs) to form the large latent complex (LLC), which is secreted into the extracellular space. LTBPs facilitate the anchorage of TGF-β to matrix proteins such as fibronectin, thereby allowing its latent storage within the ECM. In response to external stimuli, TGF-β activation can be spatially and temporally regulated to exert its fibrogenic and immunomodulatory functions [40,41].

The release of TGF-β from its latent complex is a prerequisite for initiating downstream signaling. This activation can be triggered by a variety of extracellular stimuli such as changes in pH, mechanical stress, integrin interactions, proteolytic activity, and ROS levels. Both acidic (pH 2.5–4) and alkaline (pH 10–12) environments can facilitate the activation of latent TGF-β [42]. Integrin αVβ6 is specifically expressed in epithelial cells, while αVβ8 is widely expressed in fibroblasts, macrophages, and tumor cells; both of these play important roles in activating latent TGF-β in the synovial ECM. Both integrins recognize the RGD (Arg-Gly-Asp) motif on LAP but employ different mechanisms for TGF-β activation. Integrin αVβ6 recognizes and binds to the RGD motif on LAP, utilizing the dynamic connection between the β6 subunit’s cytoplasmic tail and the actin cytoskeleton to create a force-transmitting bridge. Mechanical deformation generated by cell migration or contraction is hence transmitted to the LLC, inducing the release of the active TGF-β cytokine [43]. In contrast, after binding to the RGD motif, integrin αVβ8 relies on the proteolytic activity of MMP14 to cleave LAP and release TGF-β [44]. A similar mechanism exists in tumor tissues, whereby Q. Yu et al. showed that MMP-9, localized to the cell surface through the hyaluronic acid receptor CD44, can cleave latent TGF-β and lead to activation [45]. These findings highlight the importance of protease-dependent TGF-β activation in tissue remodeling and disease, including in the context of OA.

The TGF-β receptors are categorized into type I and type II, both of which possess intracellular domains that link serine/threonine kinase and tyrosine kinase activities. Although lacking a typical kinase domain or intracellular effector module to directly initiate the TGF-β signaling cascade, the auxiliary receptor TβRIII compensates functionally through its high-affinity ligand-binding properties, particularly through its preferential recognition of TGF-β2. TβRIII acts as a co-receptor that facilitates the presentation of TGF-β ligands to the TβRII/TβRI signaling complex, thereby enhancing ligand capture efficiency and sensitizing the receptor system to lower concentrations of TGF-β. In FLSs, active free TGF-β in the ECM binds first to TβRII, which subsequently recruits and phosphorylates the type I TGF-β receptor. Among type I receptors, the two key variants activin receptor-like kinase 1 (ALK1) and ALK5 mediate different signaling outcomes. Upon activation, ALK5 initiates the canonical Smad-dependent TGF-β signaling pathway. After TGF-β1 binds to its receptor, it activates the phosphorylation of Smad2/3, forming a Smad2/3–Smad4 complex that enters the nucleus to regulate the transcription of myofibroblast marker genes such as α-SMA and COL1A1 [46]. In parallel, ALK5 can also activate non-canonical pathways not based on Smad proteins, including RhoA-ROCK, PI3K-AKT/mTOR, and MAPK cascades such as p38, JNK, and ERK1/2. In contrast, ALK1 activates non-canonical Smad-based signaling by phosphorylating Smad1/5/8 [47]. Subsequently, Smad1, Smad5, and Smad8 associate with Smad4 and enter the nucleus as a tetrameric complex to regulate gene expression, although with transcriptional targets distinct from the ALK5-mediated pathway [14]. The dynamic balance between ALK1 and ALK5 signaling contributes to the context-dependent outcomes of TGF-β signaling in synovial fibrosis.

TGF-β regulates a broad spectrum of gene expressions in OA-FLS that define the myofibroblast phenotype, hence playing a central role in fibrogenesis [5]. Key TGF-β target genes include the following (Figure 2): (1) ACTA2 encoding α-SMA, a hallmark of myofibroblasts and a core structural component of actin stress fibers. (2) Genes encoding ECM components involved in fibrosis, such as COL1A1, COL3A1, COL5A1, and FN1, which contribute to excessive ECM accumulation. (3) Genes encoding enzymes for collagen modification and crosslinking, including PLOD, P4HB, and LEPRE1 (involved in lysine hydroxylation during collagen synthesis), with LOX as well as P4HA3 (mediating ECM maturation and stability). In particular, the enzyme LH2, encoded by PLOD2, catalyzes the hydroxylation of lysine prior to collagen secretion. In the ECM, LOX induces crosslinking of collagen molecules containing hydroxylysine residues, thereby enhancing their resistance to protease degradation and hence matrix stiffness. (4) Genes for TIMPs and ECM-stabilizing molecules such as PRG4, which collectively inhibit matrix degradation and reinforce fibrotic remodeling. In support of the conclusions above, Remst et al. [48] confirmed that under TGF-β stimulation, OA-FLS proliferation and migration was enhanced in the mouse synovium, accompanied by upregulated expression of collagens type I/III/IV/V, fibronectin, PLOD2, LOX, and TIMP. Thereby, TGF-β signaling becomes a driving force for OA-FLS to transit into a fibrotic phenotype, fostering synovial fibrosis.

The negative regulation of the TGF-β signaling pathway is primarily mediated by inhibitory Smads (I-Smads), namely Smad6 and Smad7. These I-Smads compete with receptor-regulated Smads (R-Smads) such as Smad 2/3 for binding to ALK, thereby preventing R-Smad phosphorylation and activation. Additionally, I-Smads can recruit Smurf proteins that tag TGFβR1 for degradation through the ubiquitin–proteasome pathway. Furthermore, I-Smads can induce dephosphorylation of activated R-Smads, causing them to dissociate from the Smad trimeric complex and suppressing downstream transcriptional activity. These inhibitory mechanisms serve as important feedback controls to fine-tune TGF-β signaling and prevent excessive fibrogenic responses. Targeting these negative regulators represents a promising molecular strategy to suppress fibroblast activation and synovial fibrosis in OA [49,50].

#### 2.2.2. Wnt/β-Catenin Pathway

The mammalian Wnt family comprises 19 secreted proteins that exhibit both functional specificity and commonality. They primarily act as intercellular signals through autocrine and paracrine mechanisms, regulating key physiological processes such as cell proliferation, differentiation, and migration; cell cycle progression; and cell polarity. In the synovium, Wnt signaling plays a critical role in maintaining tissue homeostasis and regulating FLS behavior. Dysregulation of Wnt pathways has been implicated in the pathogenesis of OA, contributing to synovial hyperplasia, inflammation, and aberrant tissue remodeling [51].

The canonical Wnt signaling pathway is activated by ligands such as Wnt3a, Wnt1, and Wnt8 through their binding to Frizzled receptors and the co-receptors LRP5/6. This interaction leads to the assembly of a receptor complex that recruits the cytoplasmic APC/Axin/GSK-3β degradation complex and inhibits its action on β-catenin. As a co-activator that can upregulate the transcription of downstream target genes [52], β-catenin is typically associated with cadherins in the cell membrane and is anchored to the actin cytoskeleton [53]. Under basal conditions, GSK-3β phosphorylates β-catenin, targeting it for ubiquitination and subsequent degradation through the ubiquitin–proteasome pathway. Inhibition of the APC/Axin/GSK-3β complex hence leads to the stabilization and cytoplasmic accumulation of β-catenin, which then translocates to the nucleus. There, β-catenin binds to TCF/LEF family transcription factors to form a complex, thereby activating the transcription of Wnt target genes [54,55,56].

In contrast, the non-canonical Wnt pathways are activated by ligands including Wnt5a and Wnt11 acting through Frizzled receptors and co-receptors like ROR2. These pathways do not rely on β-catenin and are instead categorized into at least two types: one type involves intracellular Ca^2+^ release, which activates protein kinase C (PKC) and calcium/calmodulin-dependent kinase II (CaMKII); the other type directly activates the c-Jun N-terminal kinase (JNK) signaling cascade. Both pathways ultimately influence gene expression by modulating the activity and nuclear localization of specific transcription factors. Notably, some ligands such as Wnt3a exhibit cross-pathway regulatory properties by engaging both canonical and non-canonical signaling routes, reflecting the complexity and context-dependent nature of the Wnt signaling network [57] (Figure 3).

Compared with normal joints, the interaction between Wnt ligands and Frizzled/LRP5/6 receptors is markedly enhanced in OA chondrocytes and synovial cells, leading to increased activation of the canonical Wnt/β-catenin signaling pathway [53]. While the precise downstream genetic mechanisms by which Wnt signaling drives fibroblast activation remain to be fully elucidated, current evidence supports its pivotal role in promoting synovial fibrosis. Based on the study by Lietman et al., the Wnt inhibitor XAV-939 has been shown to suppress synovial fibroblast proliferation and type I collagen expression in a murine model of experimental OA, thereby alleviating synovial fibrosis [58]. Research by Liao et al. showcase that by repressing Wnt/β-catenin signaling in FLSs, low-intensity pulsed ultrasound successfully attenuated the expression of fibrotic genes covering α-SMA, CTGF, and Col I in a DMM mouse model of OA [59]. Insights from other tissues have revealed potential roles of Wnt target genes, including Cyclin D1, c-Myc, and MMPs, whose upregulation is implicated in fibroblast activation leading to increased proliferation, migration, and ECM remodeling, thereby promoting fibrosis [60,61]. In addition, Wnt signaling can interact with other fibrotic pathways. For example, it can synergize with the TGF-β pathway to amplify fibrotic responses in synovial fibroblasts [62]. Moreover, Wnt activation may facilitate fibrosis by inducing cellular phenotype transformation such as epithelial–mesenchymal transition (EMT), thereby endowing fibroblasts with enhanced migratory and contractile properties that increase their fibrotic capacity [63] (Figure 3).

#### 2.2.3. Hippo-YAP/TAZ Pathway

The Hippo pathway is highly conserved in mammals and is a canonical signaling pathway closely involved in controlling cell proliferation, apoptosis, and differentiation. Through these functions, it governs organ size, tissue regeneration, and injury repair. Signal transmission in the Hippo pathway primarily occurs through kinase cascades and phosphorylation events. Activation begins with mammalian STE20-like kinases 1/2 (MST1/2), which are phosphorylated either through autophosphorylation or through upstream kinases such as TAO kinases (TAOK1/2/3) or neurofibromin 2 (NF2). Phosphorylated MST1/2 binds to Salvador homolog 1 (SAV1), forming a complex that phosphorylates and activates the large tumor suppressor kinases 1/2 (LATS1/2). Subsequently, activated LATS1/2 forms a complex with MOB1A/B and phosphorylates the downstream effectors Yes-associated protein 1 (YAP) and WW domain-containing transcription regulator 1 (TAZ). Phosphorylated YAP/TAZ create docking sites for 14-3-3 proteins, which sequester YAP/TAZ in the cytoplasm and prevent their nuclear translocation [64,65]. In the cytoplasm, YAP/TAZ are either retained or targeted for ubiquitination and proteasomal degradation, effectively silencing their transcriptional activity [66] (Figure 4).

YAP/TAZ, the key downstream effector of the Hippo pathway, can significantly upregulate the expression of genes that promote cell growth and survival, such as connective tissue growth factor (CTGF) [66,67]. This transcriptional activity makes a key contribution to fibroblast proliferation, differentiation into myofibroblasts, and resistance to apoptosis. In the context of OA, the Hippo pathway serves as a crucial regulator of FLS activity. Specifically, phosphorylated YAP induces G2 cell cycle arrest in OA-FLS, effectively inhibiting FLS proliferation and reducing the pool of myofibroblast precursors [68]. CTGF, a well-established mediator of synovial fibrosis, enhances FLS proliferation, migration, chemokine synthesis, stress fiber formation, and ECM deposition by modulating cell surface proteoglycans [69]. In vivo studies have shown that adenoviral overexpression of human CTGF in mouse synovial lining led to upregulated expression of type I and III collagen and TIMP1 within 21 days [70], confirming the role of CTGF in promoting synovial fibrosis, although the effect may be transient. Additionally, CTGF can upregulate TGF-β expression and downregulate Smad7 as a negative regulator of TGF-β signaling [71], thereby sustaining FLS activation and fibrotic responses through the TGF-β pathway.

Beyond their direct effects, YAP/TAZ also interact with other signaling pathways to regulate fibrosis. Nuclear-localized YAP can bind to β-catenin, increasing its nuclear accumulation and amplifying Wnt signaling, thereby promoting FLS activation [72]. In contrast, phosphorylated (inactivated) YAP/TAZ downregulate Wnt signaling by inhibiting DVL2 phosphorylation [73] and promoting β-catenin degradation through the β-catenin destruction complex [74]. Furthermore, phosphorylated YAP/TAZ can directly modulate TGF-β signaling by (1) inhibiting Smad2/3 phosphorylation to block downstream responses of the TGF-β pathway [66] and (2) enhancing AP-1 activity, which in turn binds to the Smad7 promoter to upregulate Smad7 expression, hence reinforcing negative feedback within the TGF-β pathway [75]. Despite these identified pathways, the precise molecular mechanisms through which YAP/TAZ regulate FLS behavior and synovial fibrosis are incompletely understood and warrant further investigation.

#### 2.2.4. The Roles of Metabolic Reprogramming

Moreover, the occurrence of metabolic reprogramming during the FMT process is worth noting. This transformation requires increased nucleotide and lipid supply for heightened fibroblast proliferation, along with greater energy demands, primarily met by ATP. Substantial ATP is also consumed by fibrillar collagen synthesis, α-SMA production, and sustaining contractile stress fiber activity. Consequently, initial fibroblast activation shifts metabolism from mitochondrial oxidative phosphorylation (OXPHOS), the main ATP source, toward aerobic glycolysis (the Warburg effect). Supporting this, Alexandra et al. found that inhibiting PDKs redirects OA-FLS metabolism from glycolysis back to OXPHOS, suppressing FLS proliferation and inflammatory cytokine secretion [1]. Similarly, the overexpression of DDIT4 substantially inhibits high glucose-induced increases in OA-FLS viability, migration, and invasion and the mRNA/protein levels of IL-1β, IL-6, and TNF-α by modulating glycolysis. This subsequently alleviates chondrocyte injury in co-culture systems with OA-FLS [2]. However, current research on OA-FLS metabolic reprogramming is limited, with most studies focusing predominantly on the pro-inflammatory phenotype. Though high proliferation is involved in the beginning of FLS–myofibroblast activation, and inflammatory factors facilitate the process, these investigations lack direct evidence linking metabolic shifts to fibrotic pathogenesis.

Early activated fibroblast generally involves upregulated glutaminolysis and dysregulated lipid metabolism such as fatty acid oxidation (FAO) [14,76]. Glutamine replenishes TCA-cycle intermediates like α-ketoglutarate (α-KG) under Warburg-like conditions. Fatty acid metabolism—including synthesis, uptake, oxidation, and disposal—plays vital cellular and organ roles. Notably, FAO provides more ATP than glucose oxidation, making it essential in high-metabolism cells. Furthermore, metabolic reprogramming progresses dynamically during FMT, culminating in a distinct metabolic signature upon full myofibroblast differentiation [14]. Nonetheless, the critical gap persists: the FMT of OA-FLS requires further experimental verification.

## 3. Impact of FLS Phenotypic Changes in OA

The phenotypic transition of FLSs into myofibroblasts is a central event in the induction of synovial fibrosis [77]. As previously described, FLSs respond to activation signals through proliferation and excessive ECM deposition, leading to synovial thickening. Differentiated myofibroblasts lead to pathological alteration in ECM composition and structure through their excessive secretion of type I and III collagen, reduced production of type II collagen, and enhanced ECM protein crosslinking under the action of PLOD2. Unlike the highly organized and cartilage-specific COL2A1 found in the ECM of normal joint tissues, the COL1A1 deposited under pathological conditions exhibits disorganized structure, abnormal crosslinking, and increased stiffness. These changes lead to the formation of fibrotic masses, clinically manifesting as joint swelling and reduced mobility [9,78].

The alteration of synovium ECM components foster joint stiffness in OA simultaneously. The ECM of metazoan is mainly composed of fibrous proteins (such as collagen and elastin) that affect its tensile strength and elasticity, proteoglycans (like proteoglycans and hyaluronic acid) that grant its viscosity, and multiadhesive glycoproteins (such as fibronectin and laminin) that can bind to the former two. ECM is viscoelastic, meaning it exhibits both elastic solid and viscous liquid-like behaviors such as stress relaxation [79,80]. Through this feature, tissues are able to adjust their structures and functions dynamically under external mechanical stress—so-called viscoelastic adaptation. However, OA-synovial tissues increase in elasticity while they decline in viscosity, leading to reduced stress relaxion competence, which contributes to pathological stress retention. Meanwhile, the presence of stress fibers composed of α-SMA in myofibroblasts endows them with enhanced contractility compared with their precursor FLS cells, hence further amplifying tissue stiffness, resulting in hardened synovium that exacerbates joint stiffness and restricted movement. To be specific, the stress fibers’ excessive contraction activity brings changes in the biophysical properties of ECM, such as progressing stiffening and heightened density. Those changes boost the cellular fibrotic pathway via the integrin dependence approach, which forms a vicious spiral, deteriorating the abnormity of ECM.

Myofibroblasts also interact with synovial macrophages by secreting the extra domain A (EDA) isoform of fibronectin, which promotes the release of pro-inflammatory cytokines such as TNF-α. These cytokines are known to sensitize and activate local nociceptors, contributing to joint pain in OA [81]. Additionally, TNF-α contributes to cartilage degradation and hence, OA progression, through its action on chondrocytes, inducing a dedifferentiated phenotype characterized by reduced type II collagen and aggrecan synthesis, elevated MMP-13 expression, and accelerated cell senescence. These changes disrupt the biomechanical properties of cartilage tissue, leading to reduced elasticity and increased brittleness [82], hence weakening its compressive load-bearing function [83]. Furthermore, since cartilage tissue lacks nerves, blood vessels, and immune infiltration, the synovial fluid produced by synovial cells is crucial for maintaining cartilage homeostasis. Upon FLS activation into myofibroblasts within the synovium, their secretory function in producing hyaluronic acid and nutrients required by chondrocytes becomes impaired, leading to reduced cartilage lubrication and nutrient deprivation of chondrocytes. The decline in cartilage nourishment together with compromised mechanical protection further exacerbate cartilage erosion and OA progression [84] (Figure 5).

## 4. Corresponding Clinical Strategy

Currently, the treatment of OA is quite limited. For patients with milder symptoms, appropriate medical exercise, weight loss, and physical therapy are common interventions. Non-steroidal anti-inflammatory drugs are mainly applied for their analgesic effect. Traditional surgical interventions such as total joint replacement serve as the ultimate approach, which impose significant physical, emotional, and financial burdens on patients and healthcare systems. As our understanding of OA pathophysiology evolves, particularly the role of FLSs in synovial fibrosis and joint degeneration, novel therapeutic strategies targeting FLS activation are emerging as promising alternatives.

We hold that TGF-β is the most significant signaling pathway in FMT due to the relatively profound knowledge of its mechanisms, fibrotic-associated target genes, and engaging proteins. Although TGF-β is a well-known driver of FLS activation and fibrosis, it is also an essential regulatory molecule for maintaining chondrocyte homeostasis and promoting cartilage ECM synthesis. Therefore, therapeutic approaches need to selectively modulate TGF-β signaling to preserve its protective effects in cartilage, which to some extent limits its application. One strategy could be to target the downstream effectors of TGF-β signaling, such as the non-canonical pathway receptor ALK1 and the LH2 enzyme, to avoid disruption of normal tissue function [85]. In support of such assumption, several preclinical trials have been conducted, revealing a series of molecules that prohibit synovial fibrosis by targeting the TGF-β pathway (some also inhibit inflammatory signaling) in FLSs (Table 1).

Chrysin alleviates synovitis and fibrosis in rat osteoarthritic FLSs via PERK/TXNIP/NLRP3 signaling, while NUPR1 suppression through trifluoperazine (TFP) in FLSs attenuates synovial fibrosis through the Smad3 pathway. Inspiringly, Danggui niantong, a traditional Chinese herb complex formula, indicated the ability to attenuate pro-inflammatory and fibrotic phenotypes in TGF-β-induced FLSs with PI3k/AKT signaling pathway. Fucoidans saw a protective effect on synovial fibrosis mediated by upregulation of nitric oxide production and modulation of the TGF-β/Smad pathway. Also, prednisolone lowered the expression of α-sma protein and type III collagen as well as FLSs’ proliferation upregulated by TGF-β. Adalimumab (Anti-TNF-α) demonstrated therapeutic potential for established digital arthritis in a Phase 2 trial. Currently, a Phase 3 trial advances for lorecivivint in osteoarthritis pain relief, with uplifting news about molecular interventions of verteporfin and resolvin D1 targeting YAP in the Hippo pathway from laboratory research.

From the perspective of reducing inflammatory responses, intra-articular administration of adalimumab, a monoclonal antibody against TNF-α, has been shown to alleviate pain in patients with severe OA [92]. However, its direct inhibitory effect on FLS activation requires further verification. More recently, the small molecule lorecivivint, which inhibits CLK2 kinase and transcriptionally suppresses Wnt pathway activity, completed the Phase 3 STRIDES trial in February 2024 [57,93]. It has shown promise in alleviating OA-related pain and represents a key advancement in Wnt-targeted therapy. Admittedly, there are several controversies on the subpopulation response heterogeneity and endpoint choosing. Current trials primarily assess lorecivivint in unilateral knee osteoarthritis (OA), yet they lack data for bilateral involvement affecting over 50% of real-world patients. Patient stratification using Kellgren–Lawrence (KL) grading—a 5-point scale based on osteophytes, joint space narrowing (JSN), sclerosis, and deformity—leads to heterogeneity. KL places greater weight on osteophytes, resulting in significant JSN variability within KL grades and diverse structural damage profiles [94]. This heterogeneity may underlie inconsistent clinical outcomes: while a Phase 2 trial illustrated pronounced therapeutic effects in the full analysis set (KL2/3 patients) [95], lorecivivint showed no significant benefit in one Phase 3 trial (*n* = 513) to KL2/3 patients with 1.5–4 mm of medial JSW [96]. In another Phase 3 trial, efficacy manifested in pain and function scores only for KL2 patients [97]. These discrepancies indicate a need for multi-parameter joint assessment and subgroup classification. Emerging evidence suggests that patients with milder structural damage (larger JSW) may derive superior symptomatic benefits [98]. Furthermore, the choice of end points should be more practical. For instance, the structural endpoints may include assessment of the comprehensive profile of OA joints, highlighting the relevance to clinical symptoms. More importantly, the functional endpoints involving pain relief, which is the essence of the therapeutic goal, mainly focus on numerical rating scale (NRS) and WOMAC pain and function scores at present. There is an appeal for applying a set of complex endpoints that contain testament on patients’ ambulation ability so as to enhance the authenticity of the therapeutic effects.

Additionally, other drugs have emerged to target the Hippo pathway, such as verteporfin (VP) as a YAP inhibitor, which has been reported to attenuate chondrocyte fibrosis and cartilage ECM stiffening [91]. Given the fibrotic pathways shared between chondrocytes and FLSs, VP may also hold potential for mitigating FLS-driven synovial fibrosis, although further preclinical validation is needed. Together, with the potential to address the underlying molecular drivers of disease progression, these emerging drug therapies provide new avenues for OA treatment beyond conventional approaches.

## 5. Conclusions and Outlook

In summary, the activation of FLSs into a myofibroblast-like phenotype is not merely a consequence of OA but also a key driver in its progression. This phenotypic shift creates a self-perpetuating cycle of inflammation and tissue damage in which the cartilage, synovium, and other joint tissues deteriorate in a mutually reinforcing manner, ultimately leading to progressive loss of joint function. Despite increasing recognition of the role of FLSs in OA, current research into synovial fibrosis remains relatively underdeveloped and requires greater attention. Several critical knowledge gaps remain in our current understanding of related disease mechanisms. For instance, beyond inflammatory mediators, the precise modes of communication and direct cellular interactions among myofibroblasts and chondrocytes, sensory neurons, or vascular cells are not fully understood. We consider that this set of questions may be the most significant, as addressing them would complement the logic chain and offer an integral explanation of the role of FLSs’ transition in OA. To overcome this urgent challenge, further studies on the communication between myofibroblasts and other joint cells, especially investigations into the phenotype alterations of the latter, need to be conducted. It also remains unclear which subtypes or clinical phenotypes of OA are more susceptible to synovial fibrosis and how the activation states of FLSs and myofibroblasts fluctuate across different disease stages. Addressing these unanswered questions will help to clarify the mechanisms driving FLS–myofibroblast transition and its role in OA disease progression.

Furthermore, due to insufficient experimental evidence, unclear molecular or drug mechanisms of action, and a lack of high-quality clinical studies, anti-fibrotic drugs targeting myofibroblast transformation have not yet made sufficient progress in OA. Biological and etiological treatment and gene therapy for OA may be future directions for research and development. The inhibition of synovial fibrosis may combine such trends to circumvent traps and gain progress. When it comes to barriers, the puzzles unsolved mentioned above, especially mechanism gaps in FLS-OA interactions, may cause safety concerns and challenges in clinical trial design to confirm the effectiveness of FLS-targeted drugs, requiring awareness and practical actions. We believe that a deeper focus on synovial fibrosis will open new avenues for mechanistic insights and therapeutic intervention in OA, ultimately transforming the management of this debilitating condition.

## Figures and Tables

**Figure 1 ijms-26-07881-f001:**
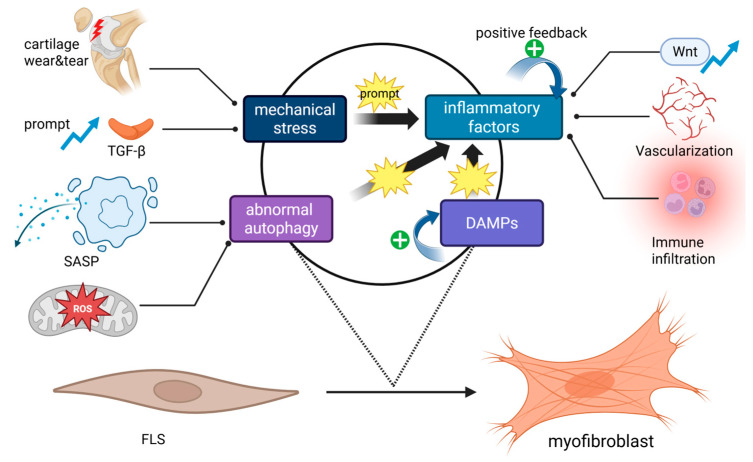
Inducing factors in the activation of FLSs into myofibroblasts. Pathological mechanical forces can promote the expression and activation of TGF-β, leading to cartilage damage and enhancing intra-articular inflammatory reactions, both of which can activate FLSs. Inflammatory cytokines (mainly interleukins) can directly induce FLS activation through the Wnt pathway while also promoting angiogenesis, immune cell infiltration, and various pro-inflammatory pathways within the cell to form a positive feedback loop. Damage-associated molecular patterns (DAMPs) also increase the levels of inflammatory cytokines through the NLRP3 inflammasome pathway. Abnormal autophagy drives the myofibroblast transition through the senescence-associated secretory phenotype (SASP) and excessive reactive oxygen species (ROS) production that mediate the TGF-β pathway (BioRender).

**Figure 2 ijms-26-07881-f002:**
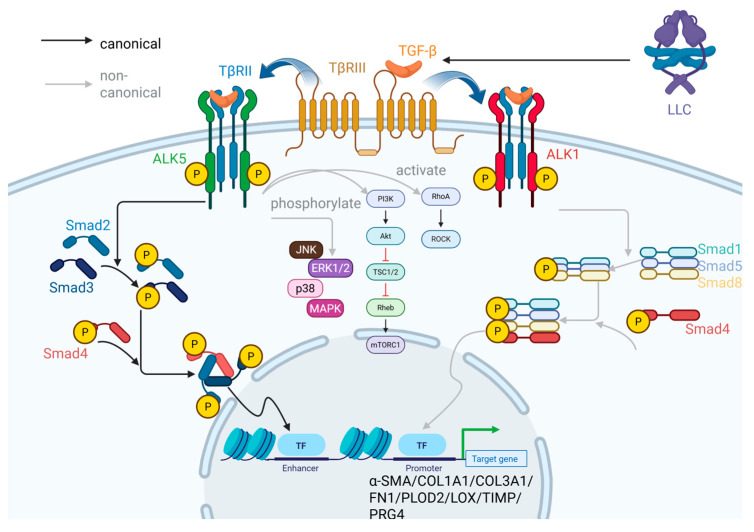
Schematic diagram of the TGF-β pathway. After TGF-β is released from the large latent complex (LLC), it binds to the receptor TβRII on FLSs to initiate a signaling cascade, which can be enhanced by the co-receptor TβRIII, leading to type I receptor activation. In the canonical pathway, activated ALK5 phosphorylates Smad2 and Smad3 proteins. Smad2/3 then bind to Smad4, and the resulting trimer translocates into the nucleus to act on transcription factors, thereby promoting the expression of fibrosis-related genes. The non-canonical pathways include the activation of p38, JNK, MAPK, and ERK1/2 pathways by ALK5, the RhoA-ROCK pathway, and the PI3K-AKT/mTOR pathway. Additionally, ALK1 phosphorylates Smad1, Smad5, and Smad8 proteins. Smad1/5/8 then bind to Smad4, and the resulting tetramer translocates into the nucleus to influence the expression of target genes (BioRender).

**Figure 3 ijms-26-07881-f003:**
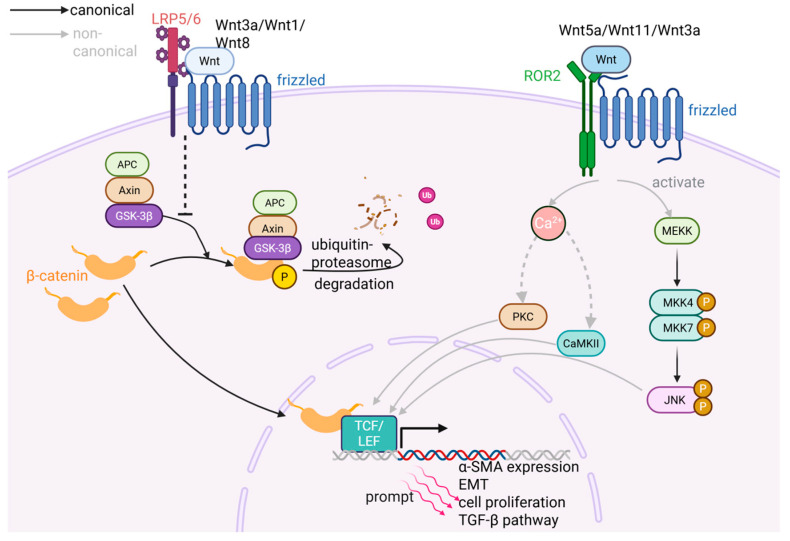
Schematic diagram of the Wnt pathway. The binding of Wnt ligands to the LRP5/6-Frizzled co-receptor activates the canonical Wnt pathway. The receptor complex binds to the APC/Axin/GSK-3β complex, thereby inhibiting its role in promoting the degradation of β-catenin. When β-catenin accumulates to a certain level in the cytoplasm, it translocates to the nucleus and acts as a co-activator to bind with the transcription factor TCF/LEF, hence promoting the expression of fibrosis-related genes (BioRender).

**Figure 4 ijms-26-07881-f004:**
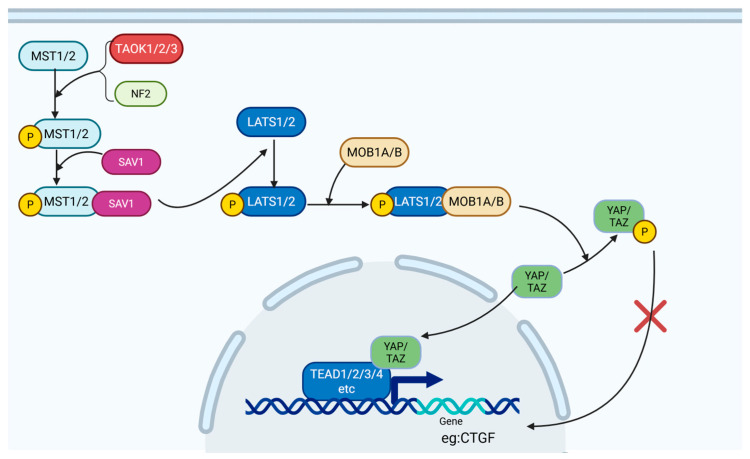
Schematic diagram of the Hippo pathway. Under Hippo signaling, MST1/2 is activated by autophosphorylation or by phosphorylation from TAOK1/2/3 or NF2. Activated MST1/2 binds to SAV1, which in turn activates LATS1/2. Phosphorylated LATS1/2 forms a complex with MOB1A/B, leading to the phosphorylation of YAP/TAZ proteins. This phosphorylation inhibits the nuclear translocation of YAP/TAZ, thereby suppressing the expression of certain fibrosis-related genes, as well as the nuclear translocation of proteins involved in other pro-fibrotic pathways (BioRender).

**Figure 5 ijms-26-07881-f005:**
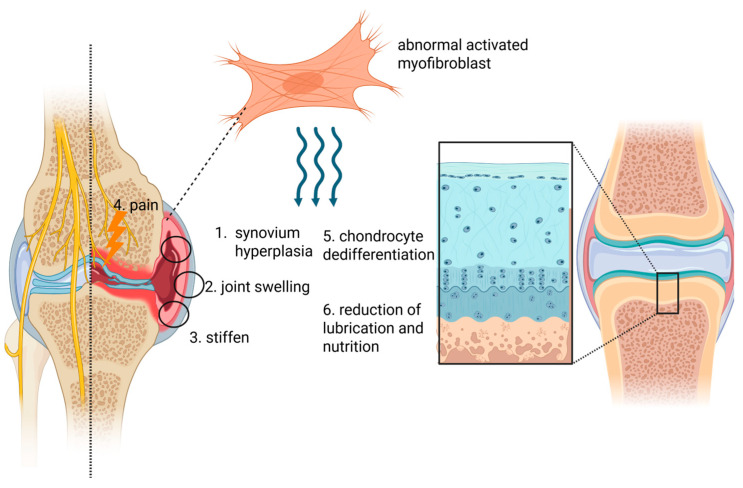
The role of FLS–myofibroblast transition in OA progression. After the activation of FLSs into myofibroblasts, their role in promoting OA is primarily manifested through (1) causing synovial hyperplasia and thickening (2) leading to joint swelling and (3) increased stiffness, (4) acting on nociceptors to induce pain, (5) inducing dedifferentiation of chondrocytes, and (6) diminishing the functions of lubrication and nutrition. These collectively result in exacerbated cartilage erosion and degeneration (BioRender).

**Table 1 ijms-26-07881-t001:** Therapeutic strategies targeting FMT and research progress.

Molecule/Drug Name	Target	Status	Preclinical/Clinical Trial Result	NCT
Chrysin	PERK/TXNIP/NLRP3 signaling	Preclinical trial	Effective [86]	\
Trifluoperazine (TFP)	NUPR1, Smad3 pathway	Preclinical trial	Effective [87]	\
Danggui Niantong	PI3k/AKT signaling, non-canonical TGF-β pathway	Preclinical trial	Effective [88]	\
Fucoidans	Nitric oxide production; TGF-β/Smad pathway	Preclinical trial	Effective [89]	\
Prednisolone	TGF-β/Smad pathway; ALK5/Smad2 signaling	Preclinical trial	Effective [90]	\
Adalimumab	TNF-α	Clinical trial, Phase 2	Completed	NCT00296894
Lorecivivint	CLK2 kinase, Wnt pathway	Clinical trial, Phase 3	Completed	NCT05603754
Verteporfin	YAP, Hippo pathway	Preclinical trial	Effective [91]	\
Resolvin D1	YAP, Hippo pathway	Preclinical trial	Effective [68]	\

## Data Availability

No new data were created in this review.

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
