# Peer review of "Fibroblast–Myofibroblast Transition in Osteoarthritis Progression: Current Insights"

_ijms, 2025, doi:10.3390/ijms26167881_

Round 1

Reviewer 1 Report

Comments and Suggestions for Authors

The reviewed manuscript presents a comprehensive and timely overview of the fibroblast–myofibroblast transition (FMT) in osteoarthritis (OA) progression, highlighting its significance as both a driver of synovial fibrosis and a potential therapeutic target. The review is well-structured, synthesizes recent advances, and identifies key molecular pathways involved in FMT. However, several areas require revision to enhance clarity, depth, and critical analysis. Specifically, the manuscript would benefit from a more balanced discussion of unresolved controversies, clearer differentiation between established findings and emerging hypotheses, and additional detail regarding translational implications for clinical practice.

Introduction

  • What is the rationale for focusing specifically on FMT, as opposed to other cellular transitions or processes, in the context of OA pathogenesis?

  • How does the author distinguish the role of synovial fibrosis in OA from its role in other joint diseases, and what evidence supports its unique contribution in OA progression?

 Results

  • What are the key differences in the activation and function of Thy1- versus Thy1+ fibroblast-like synoviocytes (FLS) during OA progression, and how do these differences influence disease outcomes?

  • How does mechanical stress interact with inflammatory signaling to modulate FLS activation and the development of synovial fibrosis in OA?

  • Which molecular pathways are most strongly implicated as therapeutic targets for inhibiting FMT, and what are the current limitations of targeting these pathways in preclinical or clinical studies?

  • What evidence supports the view that synovial fibrosis is a primary pathogenic mechanism rather than a secondary consequence in OA, and how robust are these data?

Conclusion

  • What are the major unresolved challenges or gaps in understanding the FMT process in OA that should be prioritized for future research?

  • How might the author envision translating the insights from FMT research into effective clinical interventions for OA patients, and what barriers remain to clinical implementation?

Author Response

Dear Editors and Reviewers:

Thank you for your letter and for the reviewers’ comments concerning our manuscript entitled “Fibroblast-myofibroblast transition in osteoarthritis progression: Current insights” (Manuscript ID: ijms-3753273).

Those comments are all valuable and very helpful for revising and improving our paper. We have studied comments carefully and have made correction which we hope meet with approval. Revised portion are marked in the paper with yellow highlight. The main corrections in the paper and the responds to the reviewer’s comments are as flowing:

Responses to the reviewer’s comments:

Comments 1: What is the rationale for focusing specifically on FMT, as opposed to other cellular transitions or processes, in the context of OA pathogenesis?

Response 1: Thank you for pointing this out. We have explained this question further in the text. The precise location of our complement is page 3, paragraph 2, line 118-120&130-134. The details of revision are as follows: Compared to other known cellular transitions or processes, we only have a smattering of the fibroblast myofibroblast transition (FMT) despite its significance and potential. Therefore, it’s time to specifically focus on FMT and tackle with its puzzles……Combat synovial fibrosis proved a worthy endeavor to alleviate OA progress. The po-tential interaction between FMT and other cellular changes would help reveal the dynamic process of OA, better estimating the disease. Overall, paying attention to FMT may pro-vide some novel insights on the understanding of OA.

Comments 2:How does the author distinguish the role of synovial fibrosis in OA from its role in other joint diseases, and what evidence supports its unique contribution in OA progression?

Response 2: Thank you for pointing this out. We’d like to explain that we nail down the role of synovial fibrosis specifically in OA through restricting the spectrum of references. We only search for the content of FTM in the OA field to circumvent disruptions.

Comments 3: What are the key differences in the activation and function of Thy1- versus Thy1+ fibroblast-like synoviocytes (FLS) during OA progression, and how do these differences influence disease outcomes?

Response 3: Thank you for raising this enlightening question. In response, we have further checked up relevant documents and provided a more profound depiction of FLS’s heterogeneity in page 2, paragraph 2&3, considering their genetic marks, position and functions.

Comments 4: How does mechanical stress interact with inflammatory signaling to modulate FLS activation and the development of synovial fibrosis in OA?

Response 4: Thank you for raising this enlightening question. As added in page 4, paragraph 2, line 169-173, due to excessive mechanical stress, chondrocytes and other broken cells release cartilage debris, cytokines and growth factors which anticipate in inflammatory cascade. Me-chanical stress promotes the invasive and fibrogenic phenotype of FLS both directly, as demonstrated above, and indirectly via the interaction with inflammatory signaling to perpetuate synovial inflammation

Comments 5: Which molecular pathways are most strongly implicated as therapeutic targets for inhibiting FMT, and what are the current limitations of targeting these pathways in preclinical or clinical studies?

Response 5: Thank you for pointing this out. As far as we are concerned, the TGF-β signaling pathway may be the hottest subject in drug design based on FLS’s activation which attributes to its well-established mechanism and multiple downstream proteins. However, the universality of this pathway may lead to an obstacle: researchers need to turn down TGF-β signaling(or part of it) in FLS selectively, without bringing side effects to regular activities in other cells. Corresponding views are in page 14, paragraph 2, line 555-560( a complex of former content and additional remarks): We holds that TGF-β is the most significant signaling pathway in FMT, due to the relatively profound knowledge of its mechanisms, fibrotic-associated target genes and engaging proteins. Although TGF-β is a well-known driver of FLS activation and fibrosis, it is also an essential regulatory molecule for maintaining chondrocyte homeostasis and promoting cartilage ECM synthesis. Therefore, therapeutic approaches need to selectively modulate TGF-β signaling to preserve its protective effects in cartilage, which to some extent limits its application.

Comments 6: What evidence supports the view that synovial fibrosis is a primary pathogenic mechanism rather than a secondary consequence in OA, and how robust are these data?

Response 6: Thank you for raising this enlightening question. We have, accordingly, illustrated more supportive experimental evidence in the text. As stated in page 14, passage 2, line 122-130, the inhibition of synovial fibrosis indicated relief in the damage of other OA joint tissues: Research by Yan et al. indicated that a drug specific for pulmonary fibrosis, nintedanib, remarkably mitigated articular cartilage degeneration plus reduced the quantity of M1 macrophages in OA model mouse[19]. Similarly, study by Wei et al. illustrated pirfenidone’s (an potent anti-fibrotic drug) effect on lowering articular cartilage degeneration as well as OARSI scores of the OA model rabbit, which means the progression of OA is deferred[20]. Synovial fibrosis derived from FMT plays a positive role in other OA pathological process including cartilaginous degeneration and synovial inflammation despite requirements for further explanation of concrete mechanism.

Admittedly, the data could be more robust with the improvement in administering the anti-fibrosis solely on FLS and supplement its cellular influence on other OA joint tissues.

Comments 7: What are the major unresolved challenges or gaps in understanding the FMT process in OA that should be prioritized for future research?

Response 7: Thank you for raising this enlightening question. We holds that ‘Several critical knowledge gaps remain in our current understanding of related disease mechanisms. For instance, beyond inflammatory mediators, the precise modes of communication and direct cellular interactions among myofibroblasts and chondrocytes, sensory neurons, or vascular cells are not fully understood. ’ in the original text may be the major unresolved challenges, because ‘We considered this set of question may top the most in significance, as addressing them would complement the logic chain and offer an integral explanation on the role of FLS’ transition in OA. To overcome this urgent challenge, further studies on the communica-tion between myofibroblast and other joint cells, especially investigations into the phe-notype alterations of the latter need to be conducted.’ ( page 16, passage 2,line 625-630)

Comments 8: How might the author envision translating the insights from FMT research into effective clinical interventions for OA patients, and what barriers remain to clinical implementation?

Response 8: Thank you for raising this enlightening question. We believed that Biological, etiological treatment and gene therapy for OA may be future directions for research and development, the inhibition of synovial fibrosis may combine such trends to circumvent traps and gain progress. (page16, passage 3, line 637-640). When it comes to barriers, the puzzles unsolved mentioned above, especially mechanism gaps in FLS-OA interactions may cause safety concerns and challenges in clinical trial designment to confirm the effectiveness of FLS-targeted drugs. ( page16, passage 3, line 640-643).  

Special thanks to you for your good comments.

We tried our best to improve the manuscript and made some changes in the manuscript. 

We appreciate for Editors/Reviewers’ warm work earnestly, and hope that the correction will meet with approval. Once again, thank you very much for your comments and suggestions.

Looking forward to hearing from you.

Thank you and best regards.
Yours sincerely

Corresponding Author: Dan Xing  (The former response made a mistake here. Please take the latest version. Thank you for your pardon.)

Reviewer 2 Report

Comments and Suggestions for Authors

An interesting and appealing perspective on the role of synovial fibroblast in OA occurrence and progression

Below are several point by point comments for the authors if they wish to consider

Maybe it would be good to reflect in the title (and abstract) the fact that actually authors focus on synovial fibroblasts

Introduction

Page 1 R33 and below – currently there is not direct correlation to which tissue exactly clinical manifestation corresponds. I do not think in this moment is safe to state synovial tissue in itself is the (only) one involved in pain, stiffness and especially not in the loss of mobility. Many more factors (some of them often neglected) are involved in clinical symptoms and particularities of clinical manifestations of OA

Please note that you present  Virchow  as the one having established the notion of synovial fibroblast which is not correct. Virchow  indeed coined the notion of (generally) fibroblasts . First one describing synovial tissue with a lining was Tonibee (without describing actual fibroblasts) Kölliker and Rudolf Virchow (mid-19th century): described different cell types in connective tissues, including the synovium, but did not specifically distinguish fibroblast-like synoviocytes from other stromal cells . Distinction between macrophage-like (type A) and fibroblast-like (type B) synoviocytes was made by Barland, Novikoff, and Hamerman (1962): Barland, P., Novikoff, A. B., & Hamerman, D. (1962). Electron microscopy of the human synovial membrane. The Journal of Cell Biology, 14(2), 207–220.

Please note reference 3 cited here is yet another review paper. If you cannot find the original work is better to mention the reference as such (reviewed by….)

It might be unimportant for you but incorrect information is going to flood “commonwealth knowledge ending up cited as true in short time.

The introductory chapter could be improved by adding several data regarding the burden of OA to justify why we are all suddenly interested in this otherwise decades long overlooked disease

Chapter1.  2. Maybe adding some more recent data regarding recent lineage-tracing or single-cell studies supporting the Thy1+/- subdivision would somehow update your exposure

Very nicely described α-SMA expression but  some  mention/differentiation /references  between transient and persistent myofibroblast states in OA  will be important for targeting reversibility eventyally as a therapeutic target

The chapter describing molecular pathways reads less organized with overlapping citations and extrapolations from other tissues that are not clearly resolved. (Such as inflammatory cytokines: overlapping, repetitive citations for IL-1β, TNF-α, IL-6—does not clarify their specific timelines or tissue-specific sources) DAMPs and inflammasome: are well  described would benefit from clarification on how inflammasome activation differs in OA in comparison to inflammatory rheumatoid arthritis  Autophagy subchapter lacks OA-specific studies directly linking ROS to FMT offering extrapolation from lung fibrosis models (e.g., Jain et al.)

The chapter regarding intracellular signaling pathways is very detailed and lacks overview to connect it to the OA situation that is the focus of the manuscript. Please reduce to a minimum overall presentation of TGF beta family and mechanism and only refer to impact on synovial fibroblasts in the case of OA (and eventually normal joint )

The Wnt/β-Catenin does not distinguish at all between OA and RA, this is important since the two diseases have different pathogeny, therapies and clinical manifestation

In chapter 3 a very good point is made on the modality of miofibroblasts influence ECM stifness, nociception and perhaps codnrocytes. However, the link between α-SMA fibers and joint stiffnes as mentioned in the introduction is presented oversimplistic, it  doesn’t consider viscoelastic adaptation or the role of synovial fluid, motor adaptation

Chapter 4

Actually, the ultimate solution for OA management is total joint replacement. This does not mean at all that OA management relies solely on it. Please revise this chapter addig short information regarding actual therapies in use or tested such as biological, novel molecules with disease modifying effect .intraarticular therapy

Authors should discuss when mentioning lorecivivint current clanenges in clinical trials as well as known controversies (subpopulation response, choice of end points). \

What you be, in the authors understanding, the reason why antifibrotic agents targeting myofibroblast transition  have yet to succeed in OA despite the information they provide regarding the role in OA?

What would be the modalities to circumvent the respective challenges eventually identified?

Reviewer 3 Report

Comments and Suggestions for Authors

The review article has a clear theme and logical structure, providing a valuable summary of research on FMT in OA. By supplementing the content, the academic value and clinical significance of the article can be further enhanced.

1. The specific regulatory details of pathways such as TGF-β/Smad and Wnt/β-catenin are briefly described, without clarifying key molecules (e.g., Smad2/3, β-catenin target genes) and upstream-downstream interactions. Taking the TGF-β pathway as an example, add: "After TGF-β1 binds to its receptor, it activates the phosphorylation of Smad2/3, forming a Smad2/3-Smad4 complex that enters the nucleus to regulate the transcription of myofibroblast marker genes such as α-SMA and COL1A1." For the Wnt pathway, distinguish between the canonical (β-catenin-dependent) and non-canonical (Wnt5a/Ca²⁺) pathways and explain their specific roles in FMT.

2. The discussion on FLS heterogeneity is insufficient: only a small subset of subpopulations is mentioned, without covering other recently reported markers and functional subpopulations. Single-cell data and related literature can be referenced to supplement this section.

3. The roles of metabolic reprogramming (e.g., enhanced glycolysis, dysregulated lipid metabolism) and epigenetic regulation (e.g., DNA methylation, histone acetylation) in FMT are not mentioned, even though these are hot topics in recent OA fibrosis research.

4. The treatment strategies section needs improvement. While mentioned in the abstract, the content is not fully expanded. Supplement with preclinical/clinical research progress and challenges.

5. Some language errors are present; carefully review and revise the entire text.

6. Add a search strategy and methods section.

7. Include 1-2 figures or tables,

For example: -

Figure : Schematic diagram of the molecular mechanisms of FMT in OA (including inducing factors, signaling pathways, and effector molecules).

Table : Therapeutic strategies targeting FMT and research progress (listing drug names, targets, and preclinical/clinical results). These additions will improve the readability of the article.

Round 2

Reviewer 1 Report

Comments and Suggestions for Authors

The revised version of manuscript is sufficient for publication. 

Reviewer 2 Report

Comments and Suggestions for Authors

The authors have addressed reviewer concerns. No further comments but congratulations for the work 

Reviewer 3 Report

Comments and Suggestions for Authors

I didnt find any author response file in the system. Sorry.

Round 3

Reviewer 3 Report

Comments and Suggestions for Authors

No other suggestions.